# The Impact of Physical Activities on Cognitive Performance among Healthy Older Individuals

**DOI:** 10.3390/brainsci10060377

**Published:** 2020-06-16

**Authors:** Blanka Klimova, Radka Dostalova

**Affiliations:** 1Department of Applied Linguistics, Faculty of Informatics and Management, University of Hradec Kralove, Rokitanskeho 62, 50003 Hradec Kralove, Czech Republic; 2Department of Physical Education and Sports, Faculty of Education, University of Hradec Kralove, Rokitanskeho 62, 50003 Hradec Kralove, Czech Republic; radka.dostalova@uhk.cz

**Keywords:** physical activities, aerobic exercises, dance, cognitive functions, healthy older people, effect

## Abstract

The population is aging in developed countries. This aging process results in many changes, both physical and mental. Over the years, there has been a gradual decline in the level of cognitive functions closely related to the ageing process, which is most often connected with ageing diseases such as dementia. So far, pharmacological treatment has not yet been able to cure this neurological disorder. Health policies around the world seek to find alternative programs and strategies to help a healthy population prevent cognitive decline and prolong active life. One such strategy appears to be physical activity. The aim of this review is to discuss the impact of physical activity on cognitive performance among healthy older individuals. The methodology of this study is based on a systematic electronic literature search of available studies found in three databases: PubMed, Web of Science and Scopus. The findings suggest that any physical activity in older age seems to have a positive impact on the improvement of cognitive function. Furthermore, it appears that dancing, due to its multiple mechanisms, might have the biggest effect on the enhancement of cognitive performance in healthy older individuals. However, controlled clinical trials of physical activity intervention in older adults are rare. Therefore, further research in this area (particularly on the amount of physical activity, its intensity and type) is needed.

## 1. Introduction

Increased scientific interest in the issue of aging and quality of elder life is based on the fact of demographic aging. The senior population is increasing worldwide [1], especially in developed countries, and this trend can be expected to continue in the future. According to the World Health Organization (WHO) [2], by 2050, the world’s population of 60 and over is expected to reach two billion, up from 900 million in 2015. The aging rate is clearly increasing dramatically. In fact, the elderly population is currently at its highest level in human history. This aging process considerably influences almost all sectors of society, including labor, finances, the procurement of goods and services (such as transportation), social protection and medical care. Older people thus seem to act as significant contributors to societal development [3].

The course of the aging process results in many changes, both physical and mental. There is a gradual decline in the level of cognitive function closely connected with the ageing process [4,5]. The decline of cognitive functions may also be one of the symptoms of dementia, which is a common neurological disorder in the elderly. There are currently around 47 million people suffering from dementia in the world and this figure is expected to triple by 2050 [6]. Dementia disrupts many of the higher cortical functions, including memory, thinking, orientation, understanding, reasoning, learning, speech, and judgment. Symptoms of dementia, manifested by a decline in cognitive function, greatly reduce the self-sufficiency of older people and thus impair the overall quality of life in old age. The most common type of dementia is Alzheimer’s disease [7,8,9]. Pharmacological treatments have not yet been able to cure Alzheimer’s disease (AD) or related dementias [10]. For this reason, as stated in the study by [11] or [12], health services around the world seek to find alternative programs and strategies to help prevent cognitive decline in a healthy population and thus ensure many years in old age without neurological disorders. Although some cognitive functions gradually decline during aging, a number of studies [13,14,15] suggest that this process can be significantly influenced by regular participation in physical activity, which is one of the modifiable lifestyle factors and the forth leading risk factor for global mortality. In fact, physical inactivity is the most crucial factor of the seven lifestyle risk factors (smoking, physical activity, weight, diet, blood glucose, cholesterol, and blood pressure) for the onset of AD. For example, in the USA, the lack of physical activity resulted in 117 billion dollars in annual health care costs and about 10% of premature mortality [16]. The US Department of Health and Human Services published the federal government’s second edition of the Physical Activity Guidelines for Americans, in which it specifies the key guidelines for promoting the health and fitness of Americans through regular physical activities. For healthy older adults, these guidelines are as follows:As part of their weekly physical activity, older adults should do multicomponent physical activity that includes balance training as well as aerobic and muscle-strengthening activities.Older adults should determine their level of effort for physical activity relative to their level of fitness.Older adults with chronic conditions should understand whether and how their conditions affect their ability to do regular physical activity safely.When older adults cannot do 150 min of moderate-intensity aerobic activity a week because of chronic conditions, they should be as physically active as their abilities and conditions allow [16].

It has been suggested that regular physical exercise has a positive effect on reducing risk for or slowing down the development of dementia [17,18,19]. The people who do physical exercises on a daily basis reduce the risk of the development of AD two times more than those who stay inactive [20,21]. The findings of these research studies [22,23,24] also indicate that it is important to start with these physical activities when one is middle-aged in order to prevent the development of cognitive impairment and keep a person physically and mentally fit in later adulthood.

Although the molecular mechanism through which physical exercise might raise brain performance is not clear yet, research [25,26] reveals that physical activities may positively affect brain function and productivity. For example, Radaka et al. [27] explain that physical exercise may enhance resistance against oxidative stress, help to recover from it and maintain cognitive function. In addition, Erickson et al. [28] claim that prefrontal and hippocampal areas seem to be more affected by physical activity than other areas of the brain. They state that physical activities may influence the endogenous pharmacology of the brain to improve cognitive and emotional functions in late life. Chaddock-Heyman et al. [29] add that physical activities may reduce damage in the gray matter. In addition, physical activities contribute to the release of neurotrophic factors, enhance blood flow, cerebrovascular health and benefits glucose and lipid metabolism for the brain [30].

There are many cross-sectional and experimental studies [4,5,11,31] proving that a greater amount of physical activity is related to other positive effects, including objective and subjective health, life satisfaction affecting quality of life, increasing the proportion of active body mass, developing cardiorespiratory fitness, increasing bone strength, reducing the risk of hypertension, diabetes and other disorders of metabolism, neurological diseases, osteoporosis, cancer, and better cognitive functioning in old age. The significant positive effects of physical activity on cognitive functions were found mainly in the area of general processing speed, reaction time, attention, memory and executive functions [13,32,33]. At the same time, according to Kallus et al. [34] or Hoyer et al. [35], these functions are most influenced by advancing age. Most authors [5,13,36,37,38,39] agree that in connection with cognitive maintenance in higher age, the most appropriate physical activities seem to be aerobic exercise, walking, running, Nordic walking, swimming, cycling, virtual cycling, and dancing. Positive effects, although not so large, have even been found in strength training. Recent research [34,40] suggests that the use of a wide range of physical exercises can play a key role in preventing cognitive decline.

The aim of this review is to explore the impact of physical activities on cognitive performance among healthy older individuals, possibly to reveal which physical activity might be most beneficial, as well as to emphasize the importance of this activity as a promising approach to the delay of cognitive decline in later age. In this review, physical activity is understood as an exercise that is planned, structured, repetitive, and performed with the aim of enhancing health and fitness.

## 2. Methods

The methodology follows the Preferred Reporting Items for Systematic Reviews and Meta-Analysis (PRISMA) guidelines. The authors systematically reviewed peer-reviewed, English-written articles published in three databases: PubMed, Web of Science, Scopus, Cochrane and Database of Abstracts of Reviews of Effects (DARE). Only intervention studies (randomized controlled trials and non-randomized controlled trials) were included. The selected studies involved groups where the population had to be cognitively unimpaired at the age of 55+. Only studies involving physical activity (aerobic exercises, dancing, running, strength-endurance, exergames, or cybercycling) intervention among healthy older individuals were included. In addition, the intervention had to last at least four weeks.

Categorical searches were conducted using the following keywords: healthy aging AND physical activities AND cognition, healthy aging AND physical activities AND cognitive skills. Although the search was not limited by any time period, the oldest articles date back to the year of 2003 and ends in September 2019. The terms used were searched using AND to combine the keywords listed and using OR to remove search duplication where possible. In addition, a backward search was also performed, i.e., references of detected studies were evaluated for relevant research studies that authors might have missed during their search. In addition, a Google search was conducted in order to detect unpublished (gray) literature. Both authors individually performed an independent quality assessment of these studies. They read the articles to assess eligibility and to determine the quality, and then they agreed on the basic quality criteria (adequately described study design, participant characteristics, control conditions, outcome measures, and key findings, with special focus on statistically significant differences (Table 1)). The authors selected these basic quality criteria using the Health Evidence Quality Assessment Tool for review articles.

The primary outcome of this review was as follows: the impact of physical activities on cognitive functions among healthy older individuals.

## 3. Results

Altogether, 719 articles were identified on the basis of the keywords from the database/journal searches. The majority of the studies were detected in the database Web of Science (449 studies). Scopus provided 184 studies, and in PubMed, 86 articles were found. Another four articles were identified from other sources, usually references of the already detected articles. After removing duplicates and titles/abstracts unrelated to the research topic, 79 English-written studies remained. Of these, only 55 articles were relevant for the research topic. These studies were investigated in full and they were considered against the following inclusion and exclusion criteria. The inclusion criteria were as follows:Only studies involving a physical activity intervention among healthy older individuals were included.The subjects had to be divided into an intervention and a control group.The intervention had to last at least four weeks.The primary outcome concentrated only on the effect of the physical activity intervention on cognitive functions of healthy elderly.The subjects had to be cognitively unimpaired and at age 55+ (the age was set for 55+ years since within this paper people above 55 years old are included in the group of ‘older people’. Although this may seem illogical, as they mostly are active and working persons, their cognitive (learning) operations require approaches specific to this age group, i.e., adult learning. In addition, the age of 55 years is the starting seniors’ age in the Czech Republic).

The exclusion criteria were as follows:The study included more interventions than just a physical activity apart from the exergames or cybercycling [9,15,41,42].The subjects were of different age range [19].The subjects were cognitively impaired [12,43].The intervention period was shorter than four weeks [5,44].The primary outcome did not focus on the effect of the physical activity on cognitive functions among the healthy elderly [45,46].Study protocols [47,48], cross-sectional studies, e.g., [4], prospective cohort studies [20], and review studies, e.g., [6,32,49,50,51,52] were also excluded.

Considering the above described criteria, eight studies were eventually included in the final analysis. Figure 1 below illustrates the selection procedure.

Although the search generated enough studies, after a thorough exploration, only eight studies remained relevant for the final analysis. Four were randomized controlled studies [14,53,54,55] and four were non-randomized controlled studies [13,56,57,58]. Five studies originated on the American continent [13,14,53,55,56] and three studies were of German origin [54,57,58]. The main purpose of these studies was to explore the effect of a physical activity (particularly aerobic exercises, dancing, exergames, and cybercycling) on cognitive functions among the healthy elderly. The intervention period lasted from six weeks to 18 months. The size of the subject sample ranged from 30 to 79 healthy older individuals. In three studies [14,55,56], the control groups were passive, i.e., the participants were not involved in any activity. In the remaining studies, the control groups were active, usually doing some physical activity, or had some other activities such as education about stress, home safety, fats and carbohydrates, diabetes, cancer, osteoporosis, immunizations, self-esteem, healthy relationships, or building better memory [55]. All studies used standardized outcome measures such as physiological assessment, neurocognitive assessment, and statistical analysis.

The findings of all detected studies except one [57] revealed that physical activities (especially dancing) had a positive impact on cognitive performance (particularly attention and verbal memory) in healthy older people. In addition, these activities contributed to greater neuroplasticity, as well as a relative risk reduction in clinical progression to mild cognitive impairment (MCI). No study reported any adverse events.

The main strengths of the detected research studies are their efforts towards objective assessment. However, the limitations involve differences in methodologies. In some studies, there were small sample sizes, passive control groups, insufficient representativeness, short duration of randomized controlled trials (RCTs), a number of dropouts during the intervention, and a lack of follow-up measurements.

The results of all studies indicate that there is an association between the performed physical activities and the enhancement of cognitive functions among healthy older people, although according to the Cochrane Collaboration’s tool for assessing risk of bias, the risk of bias assessed was rather high. The studies were blinded, but there were inadequate sample sizes, insufficient representativeness, short duration of RCTs, a lack of follow-up measurements, and not all the studies were randomized.

Table 1 below provides an overview of the main findings from the selected studies. The findings are summarized in alphabetical order of their first author.

## 4. Discussion

As already mentioned, only eight studies met the pre-determined inclusion/exclusion criteria. All these independent studies had a common research intention, specifically to determine the impact of physical exercises on increasing or maintaining the existing level of cognitive functions in a healthy older population. The results indicate that countries with developed economies (USA and Germany) and experiencing the impact of recent demographic changes are highly involved in the research on preventing strategies for healthy older population groups

The findings show that there might be positive effects of physical exercise on cognitive functions, particularly on attention, verbal memory and episodic memory [53,54,55,56], regardless of the intervention and control groups. Only Maass et al. [57] demonstrated no significant relationship between physical activity and cognitive functions. The possible reason for not finding a positive effect might be the number of respondents, which is the second smallest of the selected studies. In addition, the length of experimental interventions and the length of exercise duration might have played an even greater role in this case. In only this study, the exercise duration was applied for 30 min; in all other studies, the physical activities lasted between 60 and 90 min. Thus, it can be assumed that exercise for 30 min (although the same number of training units per week) may be considered insufficient in terms of impact on cognitive functions. As Buchman et al. [59] states, a higher level of total daily physical activity is connected to a reduced risk of dementia. As Taylor [60] points out, international recommendations for physical activity and exercise in older adults consistently recommend a moderate level aerobic exercise for 30 min per day for five days of the week, combined with two days of strength training. This is in line with other researchers such as Langhammer et al. [61] who suggest that healthy older people should ideally exercise 150 min a week for at least six months. In addition, research [62] demonstrates that moderate-intensity exercise is associated with improved performance in working memory and cognitive flexibility, while high-intensity exercise enhances the speed of information processing. In this sense, it has been illustrated that peripheral brain-derived neurotrophic factor increased considerably after high intensity exercise, but not after low-intensity exercise. It has been confirmed that high-intensity exercise generates bigger benefits for cognitive performance than low-intensity exercise in older people. In addition, Ohman et al. [63] in their review on cognitive performance in older adults with mild cognitive impairment (MCI) or dementia found similar results concerning improvements in global cognition, executive function, attention and delayed recall, but only among people with MCI. This indicates that physical exercise might also have positive cognitive effects on patients with MCI.

Furthermore, it is difficult to determine which physical activity appears to be the most appropriate since the design of all studies is based on interventional programs. The study by Berryman et al. [13] shows that any type of physical activity, be it a high-intensity aerobic and strength-training program, or gross motor activities, has the same effect on the improvement in cognitive performance. Nevertheless, the comparison of effects on particular types of physical exercises appears to be quite surprising. In this context, a study by Esmail et al. [14] should be mentioned. In this study, one intervention group performed aerobic exercises and the other intervention group was enrolled in dance training, which can be classified as an aerobic activity. Although both groups showed positive impacts of exercise on cognitive functions, dance was a major influence on cognitive functions. One of the possible explanations may be that during dancing, apart from the pivotal influence on physical fitness, there was an additive effect on other structures (i.e., remaining upright, balance ability and musical accompaniment) which can influence subjects’ emotions. Similar findings are reported by Muller et al. [54] who suggest that participating in a long-term dance program that requires constant cognitive and motor learning is superior to engaging in repetitive physical exercises (i.e., strength-endurance exercise) in inducing neuroplasticity in the brains of seniors. As Klimova et al. [64] reveal, the type of dance plays an important role. Their research results show that rhythmic dancing (i.e., Salsa, Danzon or a circle dance) has a far-reaching effect on the overall performance of older healthy adults. The type of dance is also associated with challenging the mind; the more decisions one has to make, the better the improvement of cognitive abilities. Therefore, freestyle dances (i.e., Latin dances) are considered to be the best option. These findings have been confirmed by the study by Marquez et al. [55] whose participants practiced four dance styles (Merengue, Cha Cha Cha, Bachata, and Salsa) which were ordered by level of difficulty and subjects in the intervention group significantly improved their episodic memory in comparison with the control group. The decline of episodic memory in later age is namely one of the defining features of dementia. Moreover, Verghese et al. [65] maintain that people who dance four times a week might reduce the risk of dementia by 76%. These findings have been also supported by Predovan et al. [52] whose review suggested that dance can be used as a promising alternative intervention to address physical inactivity and to cognitively stimulate older adults. In addition, it can positively affect older people’s mood by its socializing effect [66], which is superior to all other forms of the mentioned physical activities.

Another interesting comparison is the study [53] where the influence of aerobic virtual cycling (cybercycling) on cognitive functions was observed. The control group also cycled, but only on a conventional stationary ergometer. The results showed that the risk of developing MCI had been reduced by 23%. Similarly, positive results related to the application of virtual physical activities can also be observed in the study by Ordung et al. [58], who conducted exergames training, which is simply computer games where a movement component is needed. Specifically, the study involved computer-aided sports disciplines (athletics, swimming, or dance); the video game is played by involving the whole body and combines a number of movement elements (coordination, strength, endurance, speed and mobility). This study showed that exergames may contribute not only to the positive impact on sensorimotor skills among the participants of the intervention group, but also to the improved performance on cognitive tasks when compared with the participants in the control group. Thus, the question arises as to why the contribution of virtual physical activity to cognitive maintenance is more important than the aerobic exercise itself. The theory proposed by Hultsch et al. [67] and Yaffe et al. [68] suggests that not only the active lifestyles in the form of various physical exercises, but also participation in cognitive activities can mitigate the negative effects of continued aging. The exergames involved in virtual physical activities (apart from the physical component) undoubtedly include the cognitive component, which may be the cause of an even more significant effect on cognitive function. In addition, according to Anderson-Hanley et al. [53], exercise involving a virtual reality/exergames support, due to its fun and motivational component, could help to ensure greater participation of exercise among older people. Another advantage is that the exercises can be set to any skill level.

In addition, Erickson and Kramer [40] report that a combination of aerobic and non-aerobic exercise has the greatest effect than training programs alone. In this respect, both Berryman et al. [13] and Anderson-Hanley et al. [53] indicate that multiple pathways may lead older people to improve cognition through different exercise programs. It should also be emphasized that most recently, multi-domain lifestyle interventions instead of a single intervention have been reported to be effective in the delay and/or prevention of cognitive impairment among healthy older people [69]. This has been also confirmed by other research studies, such as [70] or [71]. In particular, the study by Reichlen et al. [70] shows that a simultaneous aerobic exercise and cognitive training intervention can significantly enhance cognitive performance among healthy older adults. These interventions usually combine two or more domains, i.e., healthy diet, physical activities, cognitive training or vascular risk management.

Table 2 below summarizes the main benefits and limitations of physical activities for healthy older individuals.

When considering the presented conclusions, it is important to take into account certain limitations of this study which include a small research set across all seven studies, passive control groups, a short duration in some trials, insufficient representativeness of subject samples, short duration of RCTs, a number of dropouts during the intervention and a lack of follow-up measurements. In particular, the unevenness of the controlled group types may yield different effect estimates. It may also provide a confusing picture regarding absolute and relative treatment efficacy. Furthermore, our selection procedure, focusing on only healthy older individuals and controlled trials, may lead to the overestimation of the effect of the described intervention results. Nevertheless, the presented limitations may serve as suggestions for future research that would explore the relationships of physical activities and cognitive functions in longitudinal randomized controlled studies.

Overall, the findings suggest that almost any physical activities might have a positive impact on the maintenance and/or improvement of cognitive performance among healthy seniors. This can contribute to the delay of neurological disorders, such as dementia, and consequently prolong the active life of these people, as well as reduce economic costs for their care. Furthermore, it appears that dancing in particular might have the biggest effect on the enhancement of cognitive performance in healthy older individuals thanks to its multiple mechanisms. Thus, this review provides a critical understanding of the importance of physical activity for a healthy individual in the ageing process. Controlled clinical trials of physical activity interventions in older adults are rare. Therefore, further research in this area (particularly on the amount of physical activity, its intensity and type) is needed.

## Figures and Tables

**Figure 1 brainsci-10-00377-f001:**
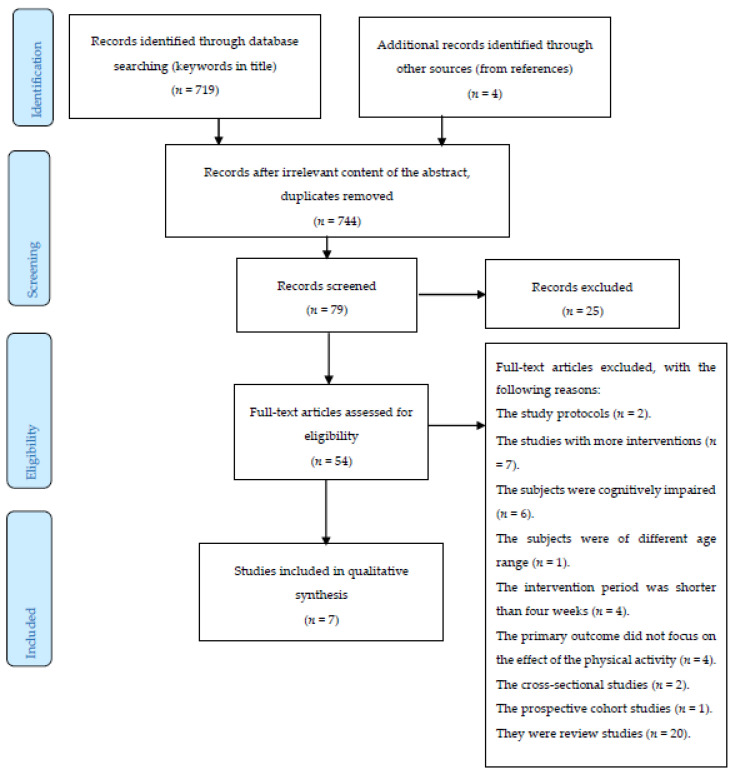
An overview of the selection procedure.

**Table 1 brainsci-10-00377-t001:** Overview of the findings from eight detected studies on the impact of physical activities on cognitive functions among healthy older individuals (authors’ own processing).

Study	Objective	Protocol Type	Number of Subjects/Population	Main Outcome Measures	Findings
Anderson-Hanley et al. [53] RCT (USA)	To investigate the effect of cybercycling on executive functions.	The intervention lasted for three months and consisted of 45 min five times a week. The intervention group was cybercycling and the control group was cycling in a traditional way.	79 healthy older adults aged 55+ years. Mean age (IG = 75.7 years, CG = 81.6 years).	Physiological assessment, neurocognitive assessment, neuroplasticity assessment, statistical analysis.	The intervention group had a 23% relative risk reduction in clinical progression to Mild Cognitive Impairment (MCI).
Berryman et al. [13] Non-RCT	To explore the impact of different training programs on the cognitive skills in healthy elderly.	Three times a week for eight weeks, 60-min session; two groups did a high-intensity aerobic and strength-training program, the third group conducted gross motor activities.	51 healthy older individuals aged between 62 to 84 years.	Physiological assessment, cognitive assessment, functional capacity tests, body composition assessment, statistical analysis.	All three groups showed equivalent improvement in cognitive performance.
Chapman et al. [56] Non-RCT (USA)	To investigate the effect of aerobic exercises on brain health of healthy elderly.	12 weeks of aerobic exercises, each lasted 60 min and was held three times a week; control group was passive.	37 healthy older individuals aged between 57 to 75 years.	Physiological assessment, neurocognitive assessment, Magnetic Resonance Imaging (MRI) scans, statistical analysis.	Even shorter intensive aerobic exercises may enhance cognitive skills and facilitate neuroplasticity among healthy older individuals.
Esmail et al. [14] RCT (Canada)	To compare the effects of dance/movement training to aerobic exercise training on cognition, physical fitness and health-related quality of life in healthy inactive elderly.	There was one intervention group for dance/movement, one for aerobic exercises and one passive control group. The intervention lasted for 12 weeks, three times a week for 60 min.	62 healthy older adults with mean age = 67.48 ± 5.37.	Physiological assessment, neurocognitive assessment, statistical analysis.	The findings showed that the dance/movement intervention but not the aerobic training had a positive impact on cognitive functions among healthy elderly.
Maass et al. [57] Non-RCT (Germany)	To explore the relationships of peripheral IGF-1, VEGF and BDNF levels to exercise-related changes in memory, hippocampal perfusion and volumes in older adults.	The subjects were divided into an aerobic exercise group (indoor treadmill, *n* = 21) and into a control group (indoor progressive-muscle relaxation/ stretching, *n* = 19). The intervention lasted for three months, three times per week, for at least 30 min.	40 healthy older individuals aged between 60 to 77 years.	Physiological assessment, neurocognitive assessment, MRI scans, statistical analysis.	IGF-I levels are linked to hippocampal volume changes and putative hippocampus-dependent memory changes that seem to occur over time independently of exercise.
Marquez et al. [55] RCT (USA)	To assess the effect of Latin dancing on cognitive function among low-active older Latinos compared to a health education control group.	The participants were divided into a dancing intervention group and an educational control group. There were 28 people in each group. The study lasted 4 months. There were two one-hour dancing lessons per week and the education program was once a week for 2 h.	57 healthy older individuals aged 55+ years.	Physiological assessment, neurocognitive assessment, statistical analysis.	The dance group showed greater improvement in episodic memory than the health education group. A main effect for time for global cognition (*p* < 0.05) was also demonstrated, with participants in both groups improving.
Muller et al. [54] RCT (Germany)	To assess whether dance training is superior to conventional strength-endurance sport activities in terms of neuroplasticity.	The participants were divided into the intervention group (dancing lessons, *n* = 26) and the control group (sport training, *n* = 26). The intervention lasted six months, 2 times for 90 min a week, and then the second phase lasted for 12 months, 90 min, once a week.	52 healthy older subjects aged 63–80 years.	Physiological assessment, neurocognitive assessment, MRI scans, statistical analysis.	After 6 months of training, the volumes in the left precentral gyrus of the dancers had increased more than those in the sport group. Both groups showed significant improvements in attention after 6 months and in verbal memory after 18 months.
Ordnung et al. [58] Non-RCT (Germany)	To examine the impact of exergame training on cognitive, motor and sensory functions in healthy elderly.	There was an intervention group doing the exergame training twice a week for 60 min for a period of six weeks, and one passive control group.	30 healthy elderly with mean age: 69.79 ± 6.34.	Physiological assessment, neurocognitive assessment, statistical analysis.	The exploratory analysis showed that the exergame training group improved in sensorimotor and cognitive tasks when compared with the passive control group.

Explanations: BDNF—brain-derived neurotrophic factor, CG—control group, IG—intervention group, IGF-1—insulin-like growth factor-I, RCT—randomized controlled study, VEGF—vascular endothelial growth factor.

**Table 2 brainsci-10-00377-t002:** Overview of the key benefits and limitations of physical activities for healthy older individuals (authors’ own processing).

Benefits	Limitations
Improvement of physical and mental health Increased fitness Improved neuroplasticity of the brain of older people Enhanced cognitive performance, especially some of the executive functions Decreased cardiovascular mortality Opportunity for social interactions Overall improvement of older people’s quality of life Non-invasive treatment Delay of aging diseases, e.g., dementia Reduction of economic burden on country’s healthcare system	A lack of randomized controlled trials Small sample sizes of people participating in the trials Short duration of the trials A lower intensity of physical exercise A lack of awareness of the importance of physical activities among people

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
