# Peer review of "The Impact of Physical Activities on Cognitive Performance among Healthy Older Individuals"

_brainsci, 2020, doi:10.3390/brainsci10060377_

Round 1

Reviewer 1 Report

The comments are below.

Comments:

  1. The first question for authors is why they believe that aerobic exercise and dance have the greatest impact on cognitive performance on elderly, see lines 19-20.
  2. Also, the authors must present in Introduction section more clear, what is the novelty of this research.
  3. Lines 110-113 - Why the authors analyzed only studies who involved PA intervention among healthy older individuals? Other studies without intervention are they considered irrelevant?
  4. From September 2019, until present, why authors don't searched articles on this topic? what is the authors explanation?
  5. AND - the authors must present this abbreviation when use for first date, in this research, also for PRISMA.
  6. Methods - this section must present more detailed in this study, is to simple presented.
  7. I did not understand why out of 54 studies only 7 remained to be analyzed on final. I think the number is to little and the authors must more detailed this selection, or, I propose to include other related studies, because is more relevant to analysis more research. If authors will do this, the review would be interesting and can fit into the topic. 
  8. Discussion- must expended with more articles because the analysis of 7 research is not enough for this review. The inclusion in article of other research will extend this chapter.

Thank you!

Author Response

Dear reviewer,

Thanks.

Authors

Reviewer 2 Report

The authors have completed a literature of experimental studies of the impact of physical exercise on cognitive function in the second half of the lifespan. It is an important topic, although the manuscript has some weaknesses, detailed below.

  1. Inaccuracies in introduction. For example: “Over the years, there has been a gradual decline in the level of cognitive function closely related to the development of dementia [3-4], which is one of the most common causes of death in later years.” The citations here are not about dementia in particular, cognitive decline in adulthood is not always associated with dementia, and dementia is not one of the top 5 causes of death for adults over age 65, although it is in the top 10.
  2. It is unclear what “it” refers to in this sentence: “It is not causal, it only allows mitigation of the course and delay the serious stages of 43 the disease.”
  3. The literature search started with 79 articles and end with only 7 studies – it appears that the authors used overly restrictive eligibility requirements. It would be interesting to see the results when less restrictive requirements were in place.
  4. Authors overstate consistency of results. According to Table 1: Berryman et al. (2014) found no differences between groups participating in quite different activities, Esmail et al. reported no impact of aerobic activities, Muller et al. reported improvements in attention and verbal memory in both experimental and control groups, and Ordnung et al did not report improvement in cognitive function. When Maass [note typo in manuscript] et al. is included, it appears that out of 7 studies, 5 showed mixed or no results.
  5. It would be appropriate to have more information about the nature of the control groups, “passive” or otherwise: did they receive social interaction at the same level of the experimental groups? In at least one study the control group appeared to be significantly older than the experimental group (Anderson-Hanley et al).
  6. Authors should not ignore that social aspect of dance in their discussion of its superiority to aerobic exercise.
  7. In discussing limitations, authors should mention selectivity of samples: only quite healthy older adults will choose to participate in a study in involving exercise and continue through the longitudinal study to the final follow-up condition. Drop-out of the less healthy participants is more likely in the experimental than control groups, which could result in overestimating the effect of the intervention.
  8. Although primarily well written, the manuscript needs to be reviewed for English language accuracy.

Author Response

Dear reviewer,

Please see the attached document about the changes.

Thanks,

Authors

Reviewer 3 Report

Dear Authors,

The present work is dealing with the effects of physical activities on cognitive functions among healthy older individuals. The systematic review is well designed and performed. I have some points for minor comments.

 General comments:

  1. Suggest adding information about any other review/systematic review of its kind (e.g. Dement Geriatr Cogn Disord 2014; 38: 347–365) and how this current review extends reader understanding of the topic. Please clarify why this review focus on healthy older individuals only? What about those with cognitive impairment?

  1. Other databases such as Cochrane and DARE databases were not checked. In addition, Clinical trial registries, such as ClinicalTrials.gov were not searched, thus trials conducted, but not published, were missed.
  2. Line 120-122: Reporting on quality assessment is seriously lacking in very important details. Please could the authors report who make the assessments, whether assessments conducted by two independent reviewers or by one reviewer and then cross-checked by another reviewer, and how disagreements would be dealt with?
  3. Was risk of bias assessed? Low and high risk scores.
  4. In terms of research quality, deciding on inclusion and exclusion criteria is not clearly defined. For example, why those at the age 55+ only? (Line 143). The intervention period was shorter than four weeks, why? (Line 150)

Author Response

(The authors gave the same response as above.)

Round 2

Reviewer 1 Report

First, the authors made some modifications but not sufficient for considerable improvement of the work.

  1. To first question the authors did not fully respond to the question, only partial. My question is not only about dancing, also about aerobic exercise. The authors must detail why specified aerobic and what was its content.
  2. To the third question I propose that the authors to include studies without intervention, it is advisable to see this research.
  3. To 4 question the authors must include in discussion section research between September 2019 and until today, this think I ask...
  4. Question no 7 - I did not understand the answer to this question 

Thank you!

Author Response

Dear reviewer,

Please see the attached document. Thank you.

Authors

Reviewer 2 Report

The authors have significantly improved their manuscript in response to reviewer comments. Introduction and Discussion sections provide a better context and interpretation for the results.

  1. In my opinion, the authors continue to overstate the consistency of the results.
  2. From the abstract: “However the studies on this research topic are rare…” It would be more accurate to say that controlled clinical trials of physical activity intervention in older adults are rare.
  3. What is the source of this information? “For example, in the 57USA, the lack of physical activity resulted in 117 billion dollars in annual health care costs and about 58 10% of premature mortality.”

  4. To state that the control groups were passive in only 3 studies is disingenuous: 3 studies is about half of your sample of 7 studies.

  5. English language concerns remain: many cases of noun-verb disagreement (at least 2 in the abstract alone)

Author Response

(The authors gave the same response as above.)
